# Examining Plogging in South Korea as a New Social Movement: From the Perspective of Claus Offe’s New Social Movement Theory

**DOI:** 10.3390/ijerph20054469

**Published:** 2023-03-02

**Authors:** Wanyoung Lee, Yoonso Choi

**Affiliations:** 1Department of Sports Industry, Hanyang University, Seoul 04763, Republic of Korea; 2Department of Sports and Health Science, Konkuk University, Chungju 27478, Republic of Korea

**Keywords:** plogging, environmental movement, trash, social movement, MZ generation, middle class, Republic of Korea

## Abstract

This study examines plogging as an environmental movement, using Claus Offe’s new social movement theory to critically analyze why its value as an environmental movement has not been recognized in Korean society. Four rounds of in-depth interviews and narrative analysis were conducted between 2 October and 28 December 2022, which involved eight individuals who participated in and organized the plogging movement. The results revealed three reasons for plogging’s failure to be appreciated by Korean society as an environmental movement: (1) the plogging movement overlaps with existing social movements; (2) the generational gap related to plogging movement participants stemming from the “new middle class”; and (3) conglomerates using the plogging movement as a marketing tool. The plogging movement has value as a new proactive, social movement for environmental protection that centers on people’s participation. However, long-standing ideological and structural issues embedded in Korean society hinder the recognition of plogging’s value.

## 1. Introduction

Plogging—a combination of “plocka up”, which means “picking up grain” in Swedish, and “jogging”, which encourages picking up trash while jogging [1]—is an emerging alternative “environmental protection” movement in Korea that can protect both people’s health and the environment. Plogging has established itself as a new trend in Korea’s MZ generation (those currently in their 20s to 40s), and has become an online cultural trend that the MZ generation uses for self-promotion [2]. By posting plogging participation on social media, these individuals package themselves as people who participate in environmental protection movements. Some media outlets suggest that the younger generation’s plogging participation originated from a value consumption trend that emphasizes environmental friendliness and public justice [3]. Plogging is more than an environmental protection movement in Korean society; it has become a part of consumption and carries value as a social building block. 

Participation in such a proactive environmental protection movement is rare in Korea. Although environmental movements in the country are not new, previous movements were not as liberal and voluntary as plogging. The environmental movement in Korean society arose from “environmental disputes” [4]. The first modern environmental dispute occurred in May 1965 in the village of Gamcheon in Busan, where 250,000 residents filed for an injunction against exhaust emissions from the Gamcheon Thermal Power Plant. Other examples include a farming family in Jeollanam-do Province who showed mercurial poisoning symptoms in 1978, which was investigated under a presidential order [5]; in December 1988, a Kori Nuclear Power Plant employee died from exposure, raising the public’s interest in pollution and contamination [6]; in July 1990, trihalomethane, a carcinogen, was detected in tap water; and in March 1991, the country experienced its largest case of phenol water contamination. Korea’s modern history includes frequent occurrences of disaster-like environmental disputes [7]. 

More recently, environmental pollution from rapidly increasing numbers of golfing facilities and the Terminal High Altitude Area Defense placement in Daegu Metropolitan City, which pollutes the river with coolants, have become major issues in the country. Environmental protection movements in Korean society are marked with disputes that aim to protect people’s right to live alongside negative effects of the country’s rapid and condensed economic development [8]. 

Plogging differs from movements that arose to fight against threats to human life in that it is a proactive, universal, and voluntary environmental movement [9], a type of proactive environmental movement that has not previously existed in the country. However, several problems have surfaced with the plogging movement’s propagation. Plogging participation is considered an ostentatious and one-off event, and has failed to develop as a universal movement. This study investigates why plogging has not been established and proliferated as a universal movement in Korean society by applying Claus Offe’s new social movement theory as a tool of analysis.

New social movements do not tackle specific political and economic issues, such as transforming society by seizing state power through labor movements, or parties based on class partisanship, or increasing bargaining power in contractual relations with capitalists [10,11]. On the contrary, new social movements focus on bringing attention and changes to issues that are marginalized or excluded from dominant paradigms, such as the environmental–ecological crisis, nuclear issues, the threat of large-scale wars, and discrimination based on gender or race [11].

Previous research has investigated various environmental movements from the perspective of new social movements. First, Mertig and Dunlap [12] noted that environmental movements represent the most successful social movements in western Europe and the United States in decades. Their study found that new social movements’ core content is environmental, and that the environmental protection movement has developed into a universal and voluntary movement with the influx of “a new social class”. Fadaee [13] examined environmental movements in Iran through new social movement theory under the premise that complete democracy is not prevalent in Iranian society. The author emphasized that the democratization of civil society is essential to realizing environmentalism, noting that Europe’s new social movements ideology cannot be wholly replicated in Iran, and that regional and historical contexts must be considered for a movement’s application. Sparrgaren and Mol [14] stated that the way people “handle” the environment contributes to the environmental crisis, and called for the need to integrate the proactiveness and solidarity components of social movement into modern society.

Plogging is often covered in sports. Shinta and Daihani [15] proposed plogging as civic education content for strengthening national resilience in Indonesia, noting the voluntary and effective nature of plogging as an environmental movement. They also emphasized developing educational content coupled with a creative movement, such as plogging, because millennials tend to find traditional civic education rather tedious. Furthermore, Vidal-Matzanke and Vidal-González [16] proposed developing hiking courses and lodging establishments in rural Spain to counter the decreasing population in those regions. They found that plogging could be developed into a tour program for introducing hiking courses. Finally, Méndez-Giménez et al. and Rozmiarek et al. [17,18] asserted that plogging is appropriate for physical education program changes resulting from the COVID-19 pandemic’s onset. They found that the pandemic shifted the education paradigm to an online, contactless, and hybrid approach; plogging acts as “hybrid content” by achieving remote engagement while also bettering one’s physical health and protecting the environment. As these studies observed, plogging adds value as an environmental movement, a physical activity scheme, and a means of education. 

Most previous studies analyzed the impact of new social movements on environmental movements. However, research is lacking that investigates social phenomena using a new social movements framework in the analysis. Additionally, few studies have examined the value of plogging as an environmental movement. This study, therefore, aimed to critically analyze why plogging’s value has not been recognized as an environmental movement in Korean society from the perspective of Offe’s [18,19] new social movements. This study provides a discussion forum to thoroughly explore the value of plogging—an activity that is appreciated worldwide for its environmental protectiveness—from an academic viewpoint. By analyzing why plogging has not taken root in Korean society, this study predicts potential problems that may arise during the initial phase of introducing the movement in the country. 

## 2. Theoretical Tool

### 2.1. New Social Movements and Offe’s Concept of New Social Movements

New social movements comprise those that emerged in the late 1960s in industrialized countries, stemming from the transition to a post-industrial society. They differ from traditional class-based social movements in that they pursue self-determination and collective identity [19]. New social movements unfolded in a social space that extended beyond the fundamental capital and labor class conflict in industrial capitalism. These social movements focus on issues that transcend class, such as the environment, anti-war and anti-nuclear agendas, women/gender issues, and civil rights [20].

New social movements reject various aspects of past social movements, such as ideologies, organizational structure, and mobilization methods. Several scholars have characterized the new social movements, including Morrow and Torresp [21], Offe [22], and Harbermas [23], which are summarized here. First, new social movements prioritize “micro-interest” to focus on daily life. Second, activists in different movements share a common set of values, such as post-materialism. Third, while traditional social movements occurred on the boundary of particular fields, classes, and regions, new social movements form private and public networks among activists, including movement organizations. Fourth, traditional social movements are characterized by being absorbed into the political system and electoral politics, whereas new social movements present themselves outside the existing political order and tend to resist established orders. Finally, new social movements are not reactionary resistance movements (e.g., national resistance movements, trade protectionism movements, or anti-foreigner movements) that have recurred throughout modern society. Instead, they conduct non-reactive and universal discussions by critically questioning the essence of capitalistic efficiency and rationality.

This study analyzes plogging participants from the perspective of new social movements. Plogging shares new social movements’ characteristics of immediacy to daily life, de-materialization, network-based reciprocity, escape from institutional politics, and recovery from the damages of capitalism [21,24]. Rather than reviewing plogging’s ideology through the new social movements framework, this study expands people’s actual participation in the activity, applying Offe’s theory, which treats participants as a key asset [22].

Offe explained new social movements by analyzing social movements’ “traditional paradigms” and the “new paradigms” pursued by new social movements based on four dimensions: agents of action, issues, values, and means of action [22]. First, he emphasized agents of action, who are defined as a new middle class that emerged outside the framework of traditional party politics and electoral politics. The new middle class recognizes social contradictions through high levels of education and economic activity. In other words, it is capable of leading the mood and social trends. Second, Offe focused on peacekeeping, the environment, human rights, and non-exclusive forms of labor, reaching beyond past issues, such as economic growth and distribution, and social control. He also challenged the limitations of everyday politics’ systems and processes within civil society on major issues. Third, Offe accentuated individual autonomy and identity as a main value, contrasting it with centralized control. The new social movement criticizes the rational system type that focuses on material progress in technology, economy, politics, and culture, and values the recovery of individual autonomy and identity as a main value. Finally, Offe distinguished between an internal and external means of action, where informality and volunteerism are internal means, and political resistance, with its negatively expressed demands, is an external means. In this respect, Offe used new social movements to cover transcendental issues, such as identity, health, neighbors, cities, the environment, and generations, and to criticize modernization.

Multiple studies have used Offe’s perspective to examine new social movements. Pichardo [25] applied Offe’s analytical framework to study social movements in western society following the emergence of today’s post-modern paradigm. The author found that the new social movements paradigm had not been critically researched in the past, and pointed to western society’s hegemony, which leads to the paradigm, as the reason behind the lack of such research. Woods [26] examined agricultural issues in rural areas through Offe’s analytical framework, and argued that rural life and culture’s traditional elements, such as agricultural reform, income levels, housing, road development, and the future of rural services, are threatened by development. Woods examined the background in which a new “rural politics” emerged and used Offe’s perspective to analyze how resistance movements are organized and operated in this context. Shakespeare [27] investigated the ways in which people with disabilities in the United States and the United Kingdom have risen as political powers over the past 20 years. The study draws a comparative analysis between the social resistance movements of people with disabilities and those of Black people, women, and homosexual people, using Offe’s framework to closely analyze the process through which socially disadvantaged people came to participate in new social movements. 

Offe’s analytical framework is often employed in new social movements’ environmental areas. Using this framework, Finger [28] asserted the need to change the role of non-governmental organizations focused on the environment. Benson [29] observed the congruence between new social movement and environmental movement concepts in British Columbia, Canada, by reviewing the social construct of environmental organizations in the province based on Offe’s [22] theory. Benson [29] reported that new social movements, such as those for peace, the environment, and women’s rights, significantly differ from common social movements in the way they use social infrastructure to attract participants, the type of problems they solve, and how they employ demonstrations. Benson stated that new social movements are a vanguard of social change to some people, but to others, they divert the attention of the bourgeois from the “real” project of liberation to other matters. Buechler [30] used Offe’s [22] analytical framework to examine collective behavior in modern society by reviewing the works of Manuel Castells, Alain Touraine, Jürgen Habermas, and Alberto Melucci—four major theorists who have contributed to the social movement paradigm. Buechler proposed a typological distinction between the “political” and “cultural” versions of the new social movement theory, and used Offe’s analytical framework to examine types of environmental protection social movements. Rather than addressing large-scale and transformative social changes, new social movements call for social change by covering relatively small-scale issues that are close to people’s daily lives, such as environmental problems and human rights for women, people of different races, and people with disabilities. Offe’s [22] analytical framework, which focuses on the agents within social movements, is a useful tool for filling the gaps that social movements may overlook.

### 2.2. Types of Plogging Participation in Korean Society

Since a trash pickup movement event hosted for the Han River in 2017, the plogging movement has gradually spread in Korea, with growing awareness of plogging [31]. Although it is hard to accurately estimate the exact number of movement participants, it is known that most are in their 20s and 30s, organize a plogging meetup through social media, and conduct plogging with anywhere from 3 up to about 30 people per event [32]. The plogging movement is primarily led by running crews, with about four events per month irregularly held in various areas of Seoul on weekdays and weekends [31]. The plogging movement is especially active in tourist destinations and the areas of Seoul, Busan, Jeju, and Incheon that have mountains and sea [3].

Four types of groups participate in Korean plogging events: individuals, social organizations, public institutions, and businesses. First, individuals who engage in plogging typically use hashtags such as “#Plogging” or “#1run1waste” in their social media posts, allowing other participants in different regions and countries to express their support; this support through social media leads to collective participation in plogging, which increases the number of people involved [3]. Searching “plogging” or “plogging campaign” on Instagram Korea reveals tens of thousands of accounts and posts related to the movement [33]. This use of social networks for plogging is a typical example of voluntary participation, and is a common adaptation of social movement in the environmental movements field.

Second, social organizations in Korea engage in plogging, of which the National Council of the Green Consumers Network (GCN) is the most active. The group strives to transition from an environmentally destructive social economic system to an environmentally friendly and sustainable social economic system through its members’ small practices [34]. With members of other environmental organizations, GCN members engage in plogging out in the streets, while also launching environmental protection campaigns. In addition to the GCN, other social organizations that utilize the environment, such as alpine, hiking, and trekking federations, also practice plogging independently or in collaboration with local communities [4].

Third, public institutions partake in plogging, with local governments hosting plogging events for citizens. Some institutions provide systematic support that allows plogging to be launched as a public event [35]. They hold plogging campaigns in the form of civil movements by teaming up with social groups or businesses. Plogging is primarily held as a citizen participation event and is usually hosted by public institutions in major cities, such as Seoul, Daegu, and Daejeon.

Finally, businesses participate in plogging through their environmental, social, and governance (ESG) activities linked to their commitment to ethical management [33]. With the recent increase in plogging among young people in their 20s and 30s, companies are also actively using exercise in their marketing efforts to capture the attention of young consumers [2]. Plogging is a useful means of achieving corporate ESG activity. Younger generations find it refreshing to see corporate owners and executives participating in plogging in the streets with their employees. Companies that are related to or directly impact the environment, such as sportswear and automobile distributors, often use plogging as an ESG activity. Outdoor apparel company The North Face’s plogging activity and “Can Crush Challenge” during its “Miracle 365 Virtual Run” event is a prime example of a company leading environmentally friendly running [36]. Furthermore, automaker Hyundai Motors collaborated with famous boyband BTS to introduce “plogging in daily life” in a video clip conveying the message “we do not wait” [37]. The vice-chairman of major Korean distribution company Shinsegae Group posted images of himself plogging around the headquarters of E-Mart, a large retail supermarket subsidiary, on Instagram [33]. The executive’s social media post served as an opportunity for the company to show the public how actively they participate in ESG activities.

Korean individuals, social organizations, public institutions, and businesses actively participate in plogging. Rather than coming together to resist the oppression of social structure, participants in the four groups autonomously control themselves, displaying “voluntary” actions to protect the environment, which may be interpreted as a new social movement. 

## 3. Methods

### 3.1. Research Design

This study examines the obstacles to plogging becoming a new social movement, and the efforts needed to establish it as a new social movement in Korean society. A narrative analysis was conducted to understand the perceptions of various groups participating in plogging; this approach was deemed most appropriate because social contexts around the spread of plogging as a social movement cannot be identified through fragmented perceptions of a handful of participants. Currently, plogging in Korea is practiced by large companies as a promotional event and is used by public institutions to promote their achievements, but it falls short of developing into a social movement. Therefore, new social movement theory was used as an analytical framework to verify participants’ and plogging exercise planners’ perceptions regarding plogging as a social movement in Korean society. The meaning was developed in a multilayered way, from the phenomenal to the culturally constructed using various public codes. Hence, it is necessary to delve deep into the topic and hermeneutically reconstruct the layered meanings through rich and persuasive methods. To this end, we posed the following questions, and prompted research participants to share their views. 

1. What is the position of the plogging movement in Korean society? This question seeks to examine how research participants view the exercise in society, where participating in environmental movements in itself is biasedly viewed as participating in social movements. It verifies the symbolic significance of plogging in Korean society by examining what led research participants to join plogging, and why they use it as an environmental movement.

2. What is the significance of plogging as a new social movement? Through this question, we aimed to identify what plogging means to those involved in environmental movements within the sphere of new social movements. It probes the significance of plogging within society by examining the phenomena that arise from plogging participation, and the personal and social values of plogging.

3. What needs to be done for plogging to establish itself as a genuine new social movement? This question considers methods that should be implemented for plogging to achieve its ultimate goal of establishing and expanding itself as a new social movement. In other words, it investigates the efforts that are required for plogging to genuinely take root as a new social movement within society, rather than being viewed as a one-off event.

Plogging participants and movement planners were selected as interview participants to overcome the limitations of text analysis. Plogging participants are more aware of its value. The plogging movement can establish itself as a new social movement when the individual perception of the environment changes and that change is extended to others. However, text analysis alone is limiting for exploring creation, meaning-making, and perception expansion.

### 3.2. Participants 

This study provided a debate forum for eight plogging participants and movement planners who shared their thoughts on the movement. From 2 October to 28 December 2022, we conducted four rounds of interviews with the eight participants, who lived in Seoul, Daejeon, and Daegu (where plogging frequently takes place). Participants were recruited through introductions made by plogging-related social networks, public institutions, and environmental organizations, and we used snowball sampling to expand the number of participants recruited from the same group. 

The most important recruitment criterion was that participants practiced plogging as a part of their daily lives and understood the plogging movement’s significance. Participants were, therefore, required to have one or more years of plogging experience, to be 20 years old or older, and to have knowledge of social movement characteristics in Korean society. We excluded individuals who participated in plogging only once or irregularly. We conducted a preliminary interview with the selected participants to provide overall study information, including the background, research problems, new social movement theory, and expected impacts of research findings. Participants also received an e-mail that summarized the movement and included a researcher-recorded audio clip on the topic. 

Before conducting the main interviews, we requested informed consent from participants via phone calls in which we provided information on the research purpose and objectives, interview methods, and the protection of personal information, then asked them if they would like to participate in the study. The main interviews were conducted individually, with each interview lasting 1.5 to 2 h.

### 3.3. Data Analysis

We conducted qualitative data analysis to allow for a deeper examination of the collected data, which comprised numerous methods, including the four-stage analysis procedure used in previous studies, where the content was checked and classified through a cyclical iterative coding and writing process. 

#### 3.3.1. Step 1—Organizing/Reading Data

In Step 1, we organized and analyzed the data collected on site, conducting content classification and organization of the preliminary survey and main interview data, which were documented using Microsoft Word. The documented file was read repeatedly, and additional explanations were added as needed to prevent arbitrary content revision. The organized file content was then verified by the interviewee. 

#### 3.3.2. Step 2—Coding

In Step 2, we coded and recorded facts learned from the collected data, verifying the categories when reading the data. During the code checking process, we recorded our thoughts to verify the processes that took place and how the data were categorized. 

#### 3.3.3. Step 3—Categorization

Step 3’s first categorization procedure was the key task of creating codes and categories from the raw data, during which we conducted the segmenting process, which entails finding and tagging expressions and words to identify and clarify the meaning of the data. Afterward, we performed segmenting on repeating words or key core words to verify their relevance to the topic. We conducted the second categorization process with a focus on confirming and linking correlations between the results of the first categorization process and categories. Finally, we analyzed correlations between the most frequently occurring codes and categorization results.

#### 3.3.4. Step 4—Analysis

We implemented the thick description approach during analysis to fully demonstrate the complexity and uniqueness of the derived categories’ content and to persuasively reorganize the content. To prevent arbitrary judgments, we reviewed the content with a professor of sports sociology and a professor of policy studies to ensure the study’s validity and reliability. Furthermore, we repeated the aforementioned analysis process with experts to continuously check whether the results interpretation persuasively answered the research questions, and to ensure the reliability of the analysis results. 

## 4. Results and Discussion

### 4.1. Plogging Overlaps with Existing Perceptions of Social Movements in Korean Society

Environmental movements in Korea emerged as civic society’s defensive mechanism against country and capital development. They were resistance movements conducted by residents whose survival was threatened by the state’s and capital’s emphasis on economic growth [34]. Therefore, environmental movements in Korean society first emerged as a “victims’ movement” for survival, rather than as people’s pursuit of a better quality of life [4].

Conversely, plogging exhibited a different pattern from previous environmental movements in Korea. Communicating through social media platforms, younger generations voluntarily participate in plogging, while businesses and public institutions plan campaigns for plogging and use it for marketing [35,36]. Such an unstandardized type of environmental movement is unprecedented in Korean society, which is why plogging cannot be markedly defined. There is no clear idea of how to approach the plogging movement, whether it should be seen as a simple event, a small practice in daily life for environmental protection, or the starting point of a larger social movement. Korean society’s lack of experience with handling environmental movements from the perspective of new social movements may contribute to ambiguity. However, the continued perception that environmental movements in Korea are social movements fighting against an irrational social structure makes it difficult to clearly define the nature of plogging. 

The expansion of the plogging movement may eventually lead to the growth of an organization with a political voice. That is how it is done in our country. In the beginning, one or two people come together to do something with pure intentions, but as more people gather, the group becomes stronger and more organized and eventually takes a political stance. Most environmental movement groups grew in this manner. (A planner for plogging at a public institution)

Local governments or environmental groups hosting plogging movements do not look particularly favorable. Plogging should be an “indifferent” exercise, where doing it or not doing it has no great significance. It should be closer to a sports activity. But when local governments or environmental groups take the lead to organize and plan plogging, the pure essence of the exercise is lost, making it look like an environmental protection activity or an exercise with a political motive, and that makes us hesitant to participate in it. (Plogging event participant living in Seoul)

The interviews showed that participants focused on “organizing” plogging, where organizing is the initial phase of political gatherings. Offe’s new social movement theory distinguishes between internal and external means of action [25], where internal means comprise movements’ informal and voluntary nature, while external means resist rational mainstream politics. In this regard, participants or those aware of plogging are hostile to injecting the external nature of political resistance into plogging movements.

This sentiment stems from Korean society’s painful history. During the past military dictatorships, the government’s political, economic, and social operations were solely focused on economic growth to achieve expedited and condensed national development, and pushed aside people’s rights, which turned them into protesting mobs. However, the people’s 29 June 1987 democratization movement was a turning point in Korean society’s social movements [5,7]. Before the democratization movement, social movements were primarily resisted by the government, but those that appeared afterward were varied civic movements [6]. Environmental movements also developed amidst this trend. The government’s 1992 attendance at the United Nation Environment Conference in Rio de Janeiro was the decisive moment when environmental movements were allowed to grow in Korean society [38]. Korea’s attendance at the conference initiated environmental movements’ expansion as civic movements. From then on, environmental movements were recognized as universal global issues that concerned people’s health and safety, rather than just as public problems [39]. 

Despite the transition, problems remain in Korea’s environmental–civic movements. Economic development prioritization and political corruption remnants systematized under 30 years of military regimes persist today, resulting in environmental movement groups becoming easily politicized [4,5]. However, these factors suggest that Korean society’s current ideology is at a turning point, moving away from prioritizing the economy toward parallel goals of economic growth and environmental protection.

Given the changing socio-cultural dynamics in Korean society and awareness among public institution plogging planners and participants in Seoul, the interviewees expressed concern that, although they began plogging with no political intentions, plogging could easily become a political movement because of Korea’s historical socio-cultural characteristics. Environmental movements in Korea are likely to overlap with other social movements because economic growth and environmental protection are perceived to grow together [7]. Research participants expressed concern about the movement being distorted into a social movement of a different nature, instead of remaining a purely environmental movement. Plogging planners could become an organized group, where the image of a particular group leading the movement overlaps with past traces of social movements. 

Given these factors, it is difficult to equate Korea’s plogging movement with Western new social movements. Since new social movements are based on the West’s mature social foundation, they may pursue a more mature human society and a better human life [7,13]. Therefore, the new social movement concept should be accepted and used in consideration of distinctive Korean characteristics. Similarly, plogging should establish itself as a civic-participation environmental movement. This would require providing diverse social opportunities for citizens to acquire the new social movement’s values. 

### 4.2. Generational Gap in Plogging Participation from the Emergence of “the New Middle Class” 

Korean people’s participation in environmental movements is still low, which is why major companies, public institutions, and social organizations tend to lead these movements [5]. Hence, these organizations are also at the center of plogging. Younger individuals in their 20s and 30s are the main plogging participants in Korea; this raises the question of why older generations’ participation is low. Offe [24] emphasizes the agents of action, particularly the participation of “the new middle class,” which can be defined as the “leading force of trends” [34]. In Korea today, trends are led by the MZ generation, a term that encompasses the millennial generation (M generation, those in their late 20s to early 40s as of 2020) and the Z generation (those in their early teens to mid-20s) [40]. Numerous firms, public institutions, and social organizations endeavor to capture the MZ generation’s interest and attention because they are at the heart of consumption and trends. Therefore, many Korean plogging-related events focus on this generation [36,37]. This creates an undesired disconnect between generations [41]. In restricting other generations’ plogging participation by focusing events exclusively on the younger generations, a generational rift arises and those excluded begin to take a negative view toward plogging. 

We are a married couple. We applied to participate in a plogging event to pick up trash while taking a stroll, but we were rejected because we were too old. The reason was absurd, and we were quite angry. (Plogging participant living in Busan)

We participated in plogging events several times. We do not know why so many events have an age restriction. Are non-MZ generationals not allowed to even take part in environment protection events? Plogging events open to only a particular generation is worse than not holding the events at all. (Plogging participant living in Daegu)

Leading a society’s trends ultimately signifies the greatest influence on political decision-making and consumer power [42]. The MZ generation has the strongest influence in Korean society [40], with the most social and cultural capital [41,43], and they actively use social media to produce and expand their social capital [44]. The MZ generation uses plogging as a form of cultural capital to expand social capital, allowing the 20- and 30-year-olds to obtain the symbolic representation of environmentally conscious intellectuals as cultural capital compared to their non-participating peers. The plogging participants who posted on their social media expanded their social capital by identifying and distinguishing themselves. 

The plogging movement, however, can also cause a generation gap as a culture of “their own”. The interviewee who wanted to participate in a Busan plogging event was rejected because they were too old, and the interviewee who participated in a plogging event in Daegu complained about organizers setting an age limit on plogging events. There should be no generational gap in environmental commitment and efforts to practice that commitment, because focusing participation on a certain generation that currently has the most influence undermines the environmental movement’s sustainability.

Nonetheless, Korean organizations that host plogging events focus on promoting participation in “the generation that represents the current times” [36]. Offe’s class can be interpreted as different generations within Korean society, where differences between generations are relatively greater than those between classes, and social movements tend to center on a particular generation rather than a particular class. Offe asserted that three groups lead new social movements: the new middle class (group of core activists and supporters in social movements), some from the old middle class (the group that simultaneously owns and works with the means of production), and those from outside the labor market and other outsiders (de-commodified groups) [45]. He particularly emphasized the new middle class as the leading force of new social movements [24], stating that their high education levels and economic activity allow them to become aware of social contradictions and take interest in new social movements [34,45]. The old middle class and outsiders, on the other hand, concentrate on recovering the damages they have experienced, and, thus, do not take much interest in social issues or matters that demand attention. 

Through Offe’s perspective of the new middle class for the MZ generation of Korean society, it is possible to understand why major companies and public institutions, such as local governments, focus on the MZ generation for plogging. The country’s college entrance rate, which describes a generation’s education level, indicates that 70% of the MZ generation enrolled in college, which is significantly higher than that of their baby boomer generation parents (30–50%) [46]. Economically, the MZ generation comprises the majority of the current workforce at companies [47]. Therefore, the MZ generation is Korean society’s middle class that influences social issues, inducing companies and public institutions to pay close attention to their interests and activities and solicit their involvement [48]. However, excessive focus on a particular generation causes a generational rift, which is apparent in plogging. 

Offe argued that new social movements should be universal to all citizens, rather than based on the demands or concerns of a particular class [45]. In other words, although new social movements are led by the new middle class, they need to aim at an “associated movement centered on the new middle class” rather than a “movement for the new middle class”. Additionally, in the sense that various powers can assimilate social movement issues as their own experience, the powers should seek a new type of movement that features diverse participants. 

### 4.3. Plogging as a Tool for Major Companies to Build an “Environmentally Friendly Image” 

Numerous companies in Korea currently participate in plogging, regardless of the nature of their business, including major department stores (Shinsegae Department Store, Seoul, Korea), supermarkets (E-Mart), financial companies (KB Financial Group, Seoul, Korea), automakers (Hyundai Motors Company, Seoul, Korea), and sports apparel manufacturers (The North Face, Seoul, Republic of Korea) [36,37,49]. Participation by companies that directly affect the environment or those that use natural resources for recreational activities is noticeable, and includes distributors (E-Mart), automakers (Hyundai Motors Company), and sports apparel manufacturers (The North Face). When companies organize plogging movements, company owners usually come out onto the street and participate in the activity with employees. However, companies do not participate in plogging with the sole intent of protecting and restoring the environment [49]. They actively promote their plogging participation appearance through mass media. Additionally, they usually organize plogging to be a “citizen participation type” event, promoting it to attract people’s participation by offering souvenirs [50]. Such cases demonstrate that companies do not view plogging as a purely environmental protection movement, but rather as a marketing strategy. Plogging event organizers who participated in the research expressed their thoughts on this situation. 

Plogging is an activity that is most participated by those in their 20s and 30s. It is important for us to make the so-called MZ generation (a term used in Korean society referring to people in their 20s and 30s) our main clients. At the same time, there is nothing more effective than plogging to build a company’s image of intellectualism that considers the environment. (Person A at the planning department of a distribution company)

All companies today are dedicated to ESG activities. In the past, companies were evaluated on quantitative financial indicators, such as the return on their investments, but now, nonfinancial measures, like the magnitude of their impact on society, have become the actual standards when evaluating their market value. And there is no better image-making tool than participating in the plogging movement to achieve this. It is a very rational means for protecting the environment, attracting consumers, and achieving nonfinancial indicators of a company. (Person A at the planning department of an automaker)

It is fulfilling to participate in plogging by coming out early in the morning to pick up trash while jogging, and also receive items of daily necessity as souvenirs… (omitted)… the host of the event will correctly dispose of the collected trash, right? (a citizen that participated in a company-hosted plogging even in Incheon)

During the interview, Person A from the planning department of a distribution company said that companies today see plogging as a highly efficient means of attracting young customers in their 20s and 30s and satisfying ESG activities [49]. As a new social movement, environmental movements value individual autonomy and identity, and criticize the rational system focused on materialistic progress [22]. However, how companies treat plogging contradicts the values asserted by new social movements; they consider it a tool for luring in potential customers and raising business indicators. In short, they strive to increase their profits through plogging. 

The new social movements’ values of autonomy and identity are lost in companies’ views of plogging. They hold plogging events solely to draw in consumers [50]. For example, plogging campaigns that offer plogging-related products, or involve plogging with pets, or are linked to donation campaigns are simply “niche marketing” through the plogging trend. Koreans currently in their 20s and 30s pay close attention to “co-prosperity development”. “Value consumption” that pursues “co-prosperity development” with the environment and society, even when purchasing a single product, is the key to opening 20- to 30-year-old Korean consumers’ wallets [51]. 

Person A in the planning department of an automaker said in the interview that the plogging movement helps corporate valuation by influencing their ESG. The ESG concept asserts that sustainable development can be achieved through transparent business management, which includes eco-friendly and socially responsible management and improved governance in corporate activities. ESG is emerging as a keyword that can determine the success or failure of a capital market or country beyond that of an individual company [52], because a company’s level of active participation in ESG activities has recently become an important indicator in corporate valuation; therefore, companies are competing to discover ESG content, and plogging is a valuable option for corporate ESG activities [53]. Consequently, plogging has become a leading ESG activity for many companies, including those in the automobile, financial, and distribution sectors, because it is emerging as an environmental movement that can help protect the environment and tackle climate change. 

Korean society has built a modern industrial and social structure through the government’s strong guidance and leadership during the past’s industrialization process. This is a typical characteristic exhibited by developing countries in Latin America and Asia [54]. Korea achieved rapid economic growth in just 30 years as a result of strict government-led development, during which large corporations played a decisive role in the accelerated growth. Korean conglomerates, such as Samsung Electronics, Hyundai Motors Company, and Shinsegae Group, worked toward economic growth like soldiers under strong government control. In short, the ultimate goal was not to prioritize corporate profits but to use corporate profit expansion to sustain the national budget [34]. This required workers who were dependent on corporations and people with consumption power; therefore, companies enacted various mechanisms to avoid any potential criticisms from workers and consumers during their profit-seeking activities [55], and plogging was a mechanism companies used to influence workers and consumers. Today, large corporations use the plogging movement as a means to control consumers, just as they did in the past. 

A major concern is that the people do not recognize the potential negative aspects of corporation-launched plogging campaigns. As an interviewee who participated in a company-hosted plogging event in Incheon mentioned in the interview, plogging participants have no clue that many of the company-provided souvenirs become another source of waste. This indicates that companies are ultimately producing more trash to promote their plogging events [56]. For instance, in November 2021, the Grand Josun Jeju hotel celebrated “World Vegan Day” by holding a plogging event where participants could trek Jeju Island’s Olle Trail. All participants were given a vegan moisturizer package from a vegan cosmetics brand, Chasin’ Rabbits. Another cosmetics brand teamed up with an organization collecting marine trash in Jeju Island, and as part of the plogging campaign, it gave out a kit to all participants that included sunscreen lotion, a crossbody bag, gloves, and a handkerchief [57]. These companies are essentially providing souvenirs that would, again, turn into trash, while participants collected trash. Additionally, there often are no plans discussed regarding post-collection plans for the trash collected during plogging (e.g., whether it would be collectively disposed of at a single place or disposed of at home by each participant). Participants should also be concerned about limited-use products distributed at these events.

Offe [22] suggested that the goal of new social movements is to criticize the rational system that is focused on materialistic progress. However, the current plogging movement in Korea, which is a representative environmental protection movement, appears to be dependent on the material advancement of large corporations, which discredits the true meaning of plogging as an environmental protection movement. Sustainable plogging movements cannot be achieved through a few one-time corporate campaigns. Therefore, corporations should consider how their products contribute to eco-friendliness, instead of investing in these event-based plogging movements. Furthermore, rather than being swayed by corporate plogging events, people should develop critical awareness regarding corporations’ illegal environmental pollution practices.

## 5. Conclusions

Using Offe’s theory as an analytical framework, this study examined the value of plogging in Korea, and critically analyzed why its value has not been wholly appreciated in society despite its role as an important environmental movement. 

It is too early for Korean society to accept and spread environmental movements’ value as new social movements, as Western societies have done, because of the lingering shadow of historical resistance-based social movements in the context of plogging in Korean society. Past environmental movements were people’s desperate struggles for survival, and ultimately, the grueling image of past environmental movements hinders the proliferation of plogging today. Second, the generational rift led by the “the new middle class” becoming the focus of plogging impedes the movement’s expansion. Offe emphasized the co-participation of all classes, with the new middle class taking a leading role in universal new social movements. However, plogging in Korean society is represented as the exclusive cultural capital of the new middle class. Finally, large corporations frequently employ plogging as a means of marketing. These companies promote themselves to the new middle class, who have significant purchasing power, by enveloping plogging with a positive corporate image. Despite being the main culprits of environmental pollution, these companies exploit plogging to avoid taking responsibility, thereby damaging plogging’s value through capitalistic behaviors. Consequently, Korea’s plogging movement is creating invisible conflicts between generations, companies, and people, which is the primary factor that prevents popularizing the plogging movement in Korean society. 

In view of this study’s findings, as a new social movement, the Korean plogging movement should work to restore its pure value, based on the simple and elemental concept of “exercising for my body while protecting the environment”. If political groups or corporate capital are involved just because people gather and form public opinions, it undermines the plogging movement’s original value and hinders efforts to develop plogging into a sustainable social movement. Too much involvement by the government or companies distorts the movement. In other words, if citizens feel that the government or companies are over-involved in the plogging movement, they may view it as a surveillance framework, rather than an environmental protection framework, which is the movement’s ultimate goal. This would make it difficult for the movement to become sustainable. Governments and companies must recognize these negative effects of excessive involvement to organize successful plogging events. 

The study findings make two significant contributions. First, they provide useful data regarding government policymaking to encourage Koreans’ voluntary and universal participation in environmental movements. Second, they provide a useful framework for analyzing Asian countries’ social and cultural structures, which make it more difficult to accept the West’s new social movements as they are, when developing approaches to implement environmental movements. In addition, this study examined the plogging movement’s value in Korean society, and more academically, applied Offe’s new social movement theory to identify the factors that prevent plogging from being fully accepted as a social movement. The findings’ implications should prompt discussions regarding how to prepare policy measures that will ensure tangible and substantial values, such as economic benefits, from a new social movement that can sustain a virtuous cycle across Korean society. 

## Data Availability

Data are not publicly available, though the data may be made available on request from the corresponding author.

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
