# Peer review of "Examining Plogging in South Korea as a New Social Movement: From the Perspective of Claus Offe’s New Social Movement Theory"

_ijerph, 2023, doi:10.3390/ijerph20054469_

Round 1

Reviewer 1 Report

Upon reviewing the findings of this study, I have come to the conclusion that the research does not provide significant enough impact in its field. Although the subject matter is intriguing and crucial, I do not perceive a pressing need to explore it further at this moment. I apologize for any inconvenience this may cause and I wish you continued success and growth in your future research endeavors.

Author Response

Dear Reviewers;

First of all, I would like to thank you for the comments and suggestions that allowed this article to greatly improve the quality of the manuscript. I agree with all your comments, and I corrected point by point the manuscript accordingly. In this document, your comments are in bold text and my responses in plain italics. As you can see the revised manuscript, I re-organized and added sentences and paragraphs based on ALL reviewers’ comments and suggestions in the manuscript since all suggestions from the reviewers should be combined with each other for the flow of information and the appropriate structure of the paper. In the revised manuscript, I highlighted in red color all the revised parts in red. Please see the manuscript along with my answers.

I would like to sincerely thank you for your advices and constructive comments.  

Reviewer 1

Q 1. Upon reviewing the findings of this study, I have come to the conclusion that the research does not provide significant enough impact in its field. Although the subject matter is intriguing and crucial, I do not perceive a pressing need to explore it further at this moment. I apologize for any inconvenience this may cause and I wish you continued success and growth in your future research endeavors.

A: Thank you so much for your comment. The author(s) have provided more discussions on each sections based on the interviews. Please see added discussions and explanations highlighted in red on the paper. 

Reviewer 2 Report

The article is very interesting and address an environmental problem in an original way. It could be improved by:

1. more detailed presentation of the results of interviews.

2. It would be really interesting to learn what professions do the interview people

Moreover, in my opinion the questions were formed in too academic way. There is no need to change questions in this research. However, in the next one it would be really useful to form questions in a way available for non-academics

Author Response

Dear Reviewers;

First of all, I would like to thank you for the comments and suggestions that allowed this article to greatly improve the quality of the manuscript. I agree with all your comments, and I corrected point by point the manuscript accordingly. In this document, your comments are in bold text and my responses in plain italics. As you can see the revised manuscript, I re-organized and added sentences and paragraphs based on ALL reviewers’ comments and suggestions in the manuscript since all suggestions from the reviewers should be combined with each other for the flow of information and the appropriate structure of the paper. In the revised manuscript, I highlighted in red color all the revised parts in red. Please see the manuscript along with my answers.

I would like to sincerely thank you for your advices and constructive comments.  

Q 1. more detailed presentation of the results of interviews.

A: Thank you so much for your comment. The author(s) have provided more discussions on each sections based on the interviews. Please see added discussions and explanations highlighted in red on the paper. 

Q 2. Moreover, in my opinion the questions were formed in too academic way. There is no need to change questions in this research. However, in the next one it would be really useful to form questions in a way available for non-academics.

A: Thank you so much for your understanding and comment. In this article, the significances based on the results have been interrogated by “Offe”’s work of new social movement as a theoretical framework, which leads the overall flow of the study to an academic way. As you commented, the author(s) have suggested that an empirical(non-academic) study on plogging movements as a follow-up research in the conclusion section. Please see on line 662-670 in red.         

Reviewer 3 Report

I have read the paper with interest.  It is a piece of social science analysis and focuses on the phenomenon of 'plogging' (picking up litter/trash while jogging) in Korea.  It adopts a new social movements framework to understand the emergence and characteristics of the practice.  It is therefore a specific study of a particular issue in a single geographical context.  It was an interesting read and I found the insights generated by the paper interesting and worth reading.

Two substantive comments are:

1) I would have liked to have seen some information on the scale of plogging in Korea.  Can anything be said about the scale of participation in the practice  (how many people go plogging, how regularly, in what sorts of places in Korea?).  Approximately when did the practice begin in Korea?  Is it more a feature of some places (e.g. large cities) rather than others?  If there is not good data, what can be said from press coverage?  Is there any way in which the reader can be given a sense of the scale and prominence of the activity?

2) In the discussion session, from line 527, the analysis seems to become normative and programmatic, suggesting what "should" and "should not" happen. (see also lines 626-631).  Until this point, the style of the paper is one of rather dispassionately investigating the plogging phenomenon as impartial social scientists.  It then switches to suggest recommendations for how plogging "should" be.  This could just be an inappropriate use of the word "should".  The authors need to be clearer about what is the purpose of the analysis.  Is it to make suggestions for how plogging might ideally develop?  Or is it just to examine the phenomenon from a new social movements perspective?

The editors may need to consider the specificity of subject matter and therefore the degree to which the paper makes a contribution to the academic literature.   

The quality of the written English is good, and there are only a couple of places where I would suggest edits to improve the sense of the paper.

line 33: delete "off "... and show that they ..."

line 34: replace "analyzed" with "suggested" 

line 68: suggest "New social movements do not deal with specific political ..."

line 530: "... the issues of social movements ..." or "... the issues of a social movement ..."

line 531: "Ultimately, ...MZ generation".  This sentence feels awkward.  What is the basis for the assertion that plogging should not become an environmental movement exclusively for the MZ generation" ?  Does this mean, if plogging is to fit with the new social movements model?  The next sentence (line 533) says the MZ generation "should unite" with other generations.  Why ?  This paragraph (lines 527-35) needs to be carefully reworked as the meaning is unclear and the basis for the claims about what should and should not happen does not currently make sense.

line 627: Similar point applies here where the paper starts to assert what companies and people "should" do

line 653: "Unapparent conflicts ... popular public movement" This sentence did not make sense to me and needs to be rewritten.  "Unapparent conflicts" and "unevolving ideologies" are awkward phrases

Author Response

Dear Reviewers;

First of all, I would like to thank you for the comments and suggestions that allowed this article to greatly improve the quality of the manuscript. I agree with all your comments, and I corrected point by point the manuscript accordingly. In this document, your comments are in bold text and my responses in plain italics. As you can see the revised manuscript, I re-organized and added sentences and paragraphs based on ALL reviewers’ comments and suggestions in the manuscript since all suggestions from the reviewers should be combined with each other for the flow of information and the appropriate structure of the paper. In the revised manuscript, I highlighted in red color all the revised parts in red. Please see the manuscript along with my answers.

I would like to sincerely thank you for your advices and constructive comments.  

Q 1. I would have liked to have seen some information on the scale of plogging in Korea.  Can anything be said about the scale of participation in the practice (how many people go plogging, how regularly, in what sorts of places in Korea?).  Approximately when did the practice begin in Korea?  Is it more a feature of some places (e.g. large cities) rather than others?  If there is not good data, what can be said from press coverage?  Is there any way in which the reader can be given a sense of the scale and prominence of the activity?

A: Thank you so much. As you commented, the scale of participation and also added an appropriate press coverage. Please see on line 200-208.

Q 2. In the discussion session, from line 527, the analysis seems to become normative and programmatic, suggesting what "should" and "should not" happen. (see also lines 626-631).  Until this point, the style of the paper is one of rather dispassionately investigating the plogging phenomenon as impartial social scientists.  It then switches to suggest recommendations for how plogging "should" be. This could just be an inappropriate use of the word "should".  The authors need to be clearer about what is the purpose of the analysis.  Is it to make suggestions for how plogging might ideally develop?  Or is it just to examine the phenomenon from a new social movements perspective?

A: Thank you. The author(s) have agreed with your comments, which there were some unclear term-usage like ‘should’ and this also makes readers feel unclear on the purpose of the study, so the author(s) have tried to make the paragraph clear (based on the purpose of the paper which aims to explore the phenomenon from a new social movement perspective) by repeatedly reading it. Please see on line 507-513.

Q 3. The quality of the written English is good, and there are only a couple of places where I would suggest edits to improve the sense of the paper.

Q 3-1. line 33: delete "off "... and show that they ..."

A: Thank you. As you requested, off has been deleted.  

Q 3-2. line 34: replace "analyzed" with "suggested" 

A: Thank you. The word has been replaced with suggested. Please see on line 30.

Q 3-3. line 68: suggest "New social movements do not deal with specific political ..."

A: Thanks. As you commented, the sentence has been revised.

Q 3-4. line 530: "... the issues of social movements ..." or "... the issues of a social movement ..."

A: Thank you. “the issues of social movement” has been revised to the issues of social movements.

Q 3-5. line 531: "Ultimately, ...MZ generation".  This sentence feels awkward.  What is the basis for the assertion that plogging should not become an environmental movement exclusively for the MZ generation" ?  Does this mean, if plogging is to fit with the new social movements model?  The next sentence (line 533) says the MZ generation "should unite" with other generations.  Why ?  This paragraph (lines 527-35) needs to be carefully reworked as the meaning is unclear and the basis for the claims about what should and should not happen does not currently make sense.

A: Thank you. The paragraph has been revised based on the purpose of this study by deleting the awkward sentences. Also, more discussions have been added each discussion section and in the conclusion section as well. Please see the added sentences in each discussion sections in red.   

Q 3-6. line 627: Similar point applies here where the paper starts to assert what companies and people "should" do

A: Thank you so much. We added some more discussions in the conclusion section. Please see on line 642-656.

Q 3-7. line 653: "Unapparent conflicts ... popular public movement" This sentence did not make sense to me and needs to be rewritten.  "Unapparent conflicts" and "unevolving ideologies" are awkward phrases

A: Thank you for your comment. The sentence has been revised added more significance.  Please see on line 637-656.